# Rapid Identification of Different Grades of Huangshan Maofeng Tea Using Ultraviolet Spectrum and Color Difference

**DOI:** 10.3390/molecules25204665

**Published:** 2020-10-13

**Authors:** Danyi Huang, Qinli Qiu, Yinmao Wang, Yu Wang, Yating Lu, Dongmei Fan, Xiaochang Wang

**Affiliations:** Tea Research Institute, Zhejiang University, # 866 Yuhangtang Road, Hangzhou 310058, China; demiwaiting@zju.edu.cn (D.H.); 21816063@zju.edu.cn (Q.Q.); 21816061@zju.edu.cn (Y.W.); 11616051@zju.edu.cn (Y.W.); ytlu@zju.edu.cn (Y.L.); 0617213@zju.edu.cn (D.F.)

**Keywords:** Huangshan Maofeng tea, ultraviolet spectrum, color difference, model, identification

## Abstract

Tea is an important beverage in humans’ daily lives. For a long time, tea grade identification relied on sensory evaluation, which requires professional knowledge, so is difficult and troublesome for laypersons. Tea chemical component detection usually involves a series of procedures and multiple steps to obtain the final results. As such, a simple, rapid, and reliable method to judge the quality of tea is needed. Here, we propose a quick method that combines ultraviolet (UV) spectra and color difference to classify tea. The operations are simple and do not involve complex pretreatment. Each method requires only a few seconds for sample detection. In this study, famous Chinese green tea, Huangshan Maofeng, was selected. The traditional detection results of tea chemical components could not be used to directly determine tea grade. Then, digital instrument methods, UV spectrometry and colorimetry, were applied. The principal component analysis (PCA) plots of the single and combined signals of these two instruments showed that samples could be arranged according to grade. The combined signal PCA plot performed better with the sample grade descending in clockwise order. For grade prediction, the random forest (RF) model produced a better effect than the support vector machine (SVM) and the SVM + RF model. In the RF model, the training and testing accuracies of the combined signal were all 1. The grades of all samples were correctly predicted. From the above, the UV spectrum combined with color difference can be used to quickly and accurately classify the grade of Huangshan Maofeng tea. This method considerably increases the convenience of tea grade identification.

## 1. Introduction

With the improvement in living standards, people have started to focus on health concepts. Many studies have reported that tea provides some apparent functions, such as preventing cancer [1], lowering the blood glucose level [2,3], and weight loss [4]. The pharmacological function of tea is attributed to its chemical components, including tea polyphenols, amino acids, caffeine, etc. These substances form via cultivation and processing. In China, there are six categories according to fermentation degree: green tea, black tea, yellow tea, white tea, oolong tea, and dark tea. Tea in the same category are distinct in appearance and flavor, having different names. Due to unique processing techniques, they have different types and contents of chemical ingredients by which the teas can be easily distinguished. However, teas with the same name may also have slight differences and may be categorized into different grades according to raw material and product quality. In this circumstance, it is difficult for consumers to distinguish the quality of tea.

Huangshan Maofeng tea, a green tea, is one of the ten famous teas in China. Six grades are used to indicate quality. The types and contents of the chemical components in different grades are quite alike. Traditionally, discrimination of tea quality was mainly dependent on chemical detection and sensory evaluation. However, the former is time-consuming and laborious. The results of sensory evaluation are easily affected by individual differences. Therefore, a fast and straightforward method is needed to objectively and fairly assess tea quality.

Many modern instruments are used in tea quality identification. Hyperspectral imaging [5,6], the electronic tongue [7,8], chemical sensor arrays [9,10], fluorescent probes [11,12], electronic noses [13], near-infrared spectroscopy [14], and gas chromatography-mass spectrometry/gas chromatography–olfactometry [15,16] have been used to classify the category, origins, and grade of tea. However, spectral image analysis and chemical sensor construction requires professional knowledge. It would be easier to judge tea quality using digital outputs. Some researchers have combined several techniques with mathematical methods to perform a multifaceted evaluation [17,18]. However, different pretreatment methods can increase the complexity, so detection is slow. The main use of tea is as a beverage, so it would be more direct and convenient to estimate tea quality from an infusion. In this study, a rapid detection method for quality discrimination based on tea infusion was constructed. Without complex pretreatment, we adopted the UV spectrum and color difference signal from the same tea infusion. The UV spectrum can provide the absorption intensity of the chemical content of tea. Color difference can digitally represent the color of tea infusions. The combination of these two methods is simple and straightforward and provides satisfactory discrimination results. By combining signal results with data analysis, tea grade can be classified. The specific steps are as follows:Chemical component detection and sensory evaluation of different grades of tea samples;Obtaining the color difference and UV spectrum data;The classification and recognition effects of single signals and combined signals combined with principal component analysis (PCA), support vector machine (SVM), random forest (RF), SVM + RF models for tea were compared.

## 2. Results

### 2.1. Traditional Detection

#### 2.1.1. Chemical Component Detection

The histogram in Figure 1 shows the average chemical component contents of different grades of tea. Here, we selected six grades of Huangshan Maofeng tea: special grade classes one, two, and three; and grades one, two, and three. We termed these samples S1, S2, S3, G1 G2, and G3, respectively. The content of water extract was G3 > G1 > G2 > S3 > S2 > S1. The content of tea polyphenols was S3 > G1 > S2 > S1 > G3 > G2. The content of amino acids was G1> G3 > G2 > S2 > S3 > S1. The content of caffeine was G1 > S2 > G2 > G3 > S1 > S3. The average contents of water extract, amino acids, and caffeine of lower-grade samples (G1–G3) were 9.72%, 18.73%, and 8.97% higher than those in higher-grade samples (S1–S3), respectively. The average content of tea polyphenols was 4.37% higher in the higher-grade samples. The phenolic ammonia ratio is the ratio of tea polyphenols to amino acids, and is used to show the coordination of tea taste. The phenolic ammonia ratio of higher-grade samples ranges from 4 to 5. For lower-grade samples, the phenolic ammonia ratio is 3 to 4. In this experiment, all samples belonged to the lower phenolic ammonia type. Generally, a low phenolic ammonia ratio means a high level of freshness, consistent with the taste characteristics of green tea. The eight catechins include epicatechin gallate (ECG), epigallocatechin gallate (EGCG), gallocatechin (GC), epigallocatechin (EGC), catechin (C), epicatechin (EC), gallocatechin gallate (GCG), and catechin gallate (CG). The highest content of the eight catechins was EGCG, followed by ECG, EGC, and EC. G3 had the highest content of six catechins (GC, GEC, C, EC, EGCG, and GCG). The contents of five catechins (GC, EGC, EC, EGCG, and GCG) were higher in lower-grade samples. The contents of the other chemical components were high in lower-grade samples and low in higher-grade samples. The higher-grade tea is made from a larger amount of tender buds and leaves so that the contents of these chemical components are lower than in lower-grade samples. 

#### 2.1.2. Sensory Evaluation

Figure 2 depicts the relationship between the five sensory factors of the six grades in tea. The score of appearance was S1 > S2 > S3 > G1 > G3 > G2. The score of liquor color was S2 > S1 > S3 > G1 = G2 > G3. The score of taste was S1 > S2 > S3 > G1 > G2 > G3. After calculation, the total score was S1 > S2 > S3 > G1 > G2 > G3. Overall, the higher the sample grade, the higher the total score. It was apparent that the taste accounted for the largest proportion of all factors, and the trend was the same for the total score. The total score of the sensory evaluation was strongly correlated with tea grade. 

### 2.2. Digital Evaluation

#### 2.2.1. UV Spectrum and Tea Grade

Figure 3a shows the UV spectrum of the six grades of Huangshan Maofeng tea samples in the spectral range of 230 to 380 nm. As shown in Figure 3a, all samples had similar spectral curves but different absorption intensities according to grade. The highest absorption peak of all samples was observed at 273 nm. This is considered the characteristic spectral characteristic of Huangshan Maofeng tea. The PCA result showed that the first principal component’s contribution was 82.25%; the second principal component’s was 16.10%. The PCA plot of the first two principal components is shown in Figure 3b. All samples were aligned linearly in the area. For samples S1 to S3, as the grade increased, the value of the PC2 axis increased and that of the PC1 axis decreased. Samples G1 to G3 had a great degree of overlap with each other. A few samples of S1, S3, G2, and G3 were discrete, above the linear region.

#### 2.2.2. Color Difference and Tea Grade

In the colorimetry results, the negative values of Δa are indicated in green and red indicates positive values. The positive value of Δb indicates the yellowness of tea liquid; the negative values are indicated in blue. The value of ΔL indicates the brightness of the liquid. ΔE and ΔE cmc represent the total color difference and ellipse ΔE formula, respectively. A three-dimensional diagram (Figure 4a), was drawn with the raw data of ΔL, Δa, and Δb. We can see that the samples of different grades can be distinguished. All samples are arranged diagonally in the space. S1 samples are in the upper-left corner, indicating that the liquid color was the brightest. The interval between S1 and the other five grade samples was large. One S2 sample was in the region of S1. G1, G2, and G3 had some overlapping samples. PCA was performed on the color with ΔL, Δa, Δb, ΔE, and ΔE cmc. The first two principal components represented 99.10% of the original variables. The distinctive effect was better than the three-dimensional figure using the original data. The S1 samples were clustered far from the other grade samples. S2 and S3 were very close, and the samples of each grade were arranged together. The discrimination of G1 to G3 is more apparent in Figure 3b. However, the distribution area of G1, G2, and G3 samples was large, and the value of the PC2 axis decreased as the grade lowered. A few samples overlapped. 

#### 2.2.3. Combined Signal and Tea Grade

The characteristics of the UV spectrum and color difference were extracted and combined. A new set of two-dimensional data was formed. The first two extracted principles represented 73.60% of the primary information. A PCA plot is shown in Figure 5. Starting with the S1 samples, the grade of the region decreased in a clockwise direction. The aggregation effect of S1, S2, and S3 samples was good. S1 is located on the far right of the PC1 axis. There are no other samples overlapping with S1. Samples areas of S2 and S3 overlap. They are located at –0.2 to 0.2 of the PC1 axis and –0.5 to –0.1 of the PC2 axis. G1 to G3 are scattered in the left area. As the grade decreased, the location of samples on the PC2 axis increased. G2 and G3 have several overlapping areas. 

#### 2.2.4. Tea Grade Prediction with Three Models

SVM, RF, and SVM combined with RF models were constructed for tea grade identification based on the UV spectrum, color difference, and combined signals. The total samples were divided into a training set and a testing set by stratified sampling. The training set accounted for 90% of the data set. To obtain more accurate classification results, we used 10-fold crosses validation to find each model’s optimum parameters. For constructing the SVM model, the penalty parameter C and kernel parameter gamma were in the range of 0 to 50. The best combination of C and gamma was based on the best classification accuracy of 10-fold crosses validation. For RF model settings, the number of decision trees was searched from 1 to 100. The best decision tree was based on the best classification accuracy. For the SVM + RF model, C and gamma ranged from 0 to 50, and decision trees were searched from 1 to 100.

The accuracies of the three models are listed in Table 1 and the optimum parameters of each model are shown in Table 2. In the SVM model, the accuracy of the single signal UV spectrum was the highest (1). However, testing accuracy was low. The single signal color difference was effective in testing, and the accuracy was 1. There were training mistakes in S2, G1, G2, and G3, and the training accuracy was 0.875. For the combined signal, training accuracy was lower than that of the UV spectrum but higher than that of the color difference. There were several training mistakes in G2 and G3. The testing accuracy was 0.875. In the RF model, all datasets had a training accuracy of 1. For the testing accuracy, only the combined signal reached an accuracy of 1. The testing accuracies of UV spectroscopy and color difference were 0.500 and 0.875, respectively. The most accurate parameters of UV spectroscopy, color difference, and the combined signal were 82, 35, and 18, respectively (Table 2). The effect of the combination model was equal to that of the RF models. The optimum parameters changed.

## 3. Discussion

The results of sensory evaluation showed that the total sensory score of higher-grade samples was higher. Generally, the higher the tea’s grade, the better the taste. In terms of the contents of chemical components, water extract, amino acids, and caffeine were higher in lower-grade samples because the raw material (buds and leaves) of high-grade Huangshan Maofeng tea are more tender. For catechins monomers, the content of EGCG was the highest in all samples. EGCG, EGC, ECG, and EC represent the main catechins in green tea [19,20]. The lower the sample grade, the higher the contents of EGCG, EC, EGC, and GCG, which is consistent with the results reported by Pan [21]. From the above, the contents of the chemical compounds can distinguish higher- or lower-grade samples. However, it was not possible to classify samples as a specific grade only based on chemical component content. 

Digital instrument detection results can be used to classify tea grade. PCA has been widely used in grade identification of tea. The PCA plot of the single signal UV Spectrum showed samples were arranged linearly by grade. However, samples of adjacent grades overlapped. The PCA plot of the single-signal color difference separated three grades of samples: S1, S2, and S3. These three higher-grade samples were located in the lower-left corner of the region. Lower-grade samples were located in the upper-right corner of the plot and had some overlaps. The PCA plot of the combined signal showed samples distributed with decreasing quality clockwise according to grade. The combined signal was more accurate than single signals. Other researchers also used these digital instruments for tea classification. Aboulwafa et al. found that the PCA plot of UV spectroscopic data can be used to separate two areas: low-quality Chinese tea, high-quality Chinese tea, and entire South Asian samples (from India and Sri Lanka) [20]. Dankowska and Kowalewski reported UV-Vis spectroscopy with PCA to identify six tea types; however, there were overlapping areas [22]. In terms of color difference, the brightness decreased with the grade of samples, which is consistent with the research results reported by Liang [23]. Lower-grade samples overlapped. Xu et al. [17] extracted 92% of the total variance using an electronic eye, and the PCA plot showed that three grades had mixed samples. Further study was needed. 

The polyphenolic compounds, glyphs, and phenolic acids are abundant in green tea, and have absorption peaks in the UV region. EGCG and ECG showed absorption peaks at 246–325 nm. Phenolic acids exemplified by gallic, chlorogenic, and caffeic acids showed absorbance maxima at 273 nm for the first and 330 nm for the last [20,24]. The color difference results displayed the color value in the red-green and yellow-blue space and the brightness of the tea infusion, providing visual data of the subtle differences in color, avoiding errors that are inherent with human observation. The combination of these two instruments can reflect the intrinsic quality of tea infusions and provide a comprehensive evaluation of tea.

In three prediction models, the combined signal performed the best. The combined signal had medium accuracy in the SVM model. Training and testing accuracy were 0.922 and 0.875, respectively. In the RF and SVM +RF models, the accuracies of the training and testing were all 1. The RF and SVM + RF models had the same training and testing accuracies with all signals, which had higher training accuracy than the SVM model. Studies of Huangshan Maofeng tea identification with models are lacking, but other green teas have been studied. Zhang et al. identified the grade of Xinyang Maojian tea (a kind of Chinese green tea) with SVM models, and the training and testing accuracies they reported were 86.11% and 87.5%, respectively [25]. Dankowska et al. proved that all the fusion data sets (synchronous fluorescence (SF) spectra + ultraviolet-visible (UV-Vis) spectra, SF + near-infrared (NIR) spectra, NIR + UV-Vis combined with the SVM method) may complement each other, having lower errors for the classification of tea type [22]. Xu et al. conducted partial least squares regression (PLSR) and used SVM and RF models to classify Xihulongjing tea, and combined electronic tongue, electronic nose, and electronic eye, which produced good effects compared with independent signals [17]. Usually, combined signals can be supplemented with single signals, and produce better results. In this study, compared to the three models’ results, the prediction effect of the RF and SVM + RF models were both good. Therefore, we think the RF model is sufficient for tea grade classification. Deng et al. used the RF model to discriminate Xihulongjing tea from other regions with an accuracy of 97.6% and correctly identified green tea from surrounding regions with an accuracy of 97.9% [26]. Wang et al., based on the joint information from the NIR and UV-Vis spectra, established a successful classification model with RF. The classification accuracy was 96% [27]. RF provides some advantages in such cases because it can deal with classification problems with unbalanced, multiclass, and small sample data without data preprocessing procedures. These results suggest that RF may be a promising pattern recognition method for tea grade identification [28]. Although more comprehensive information can be obtained by combining multiple instruments, the combination of these two instruments was sufficiently accurate and is more convenient and straightforward.

## 4. Materials and Methods 

### 4.1. Experimental Tea

Huangshan Maofeng tea, a famous Chinese green tea, was collected for the experiments in this study. All samples were from Anhui province, China. The six grades included: special grade class one, special grade class two, special grade class three, grade one, grade two, and grade three. We termed these samples S1, S2, S3, G1 G2, and G3, respectively. All samples were conserved in the refrigerator at 4 °C before the experiments.

### 4.2. Chemical Components Detection

In this experiment, 12 chemical components of tea were measured according to the National Standard of the People’s Republic of China. The water extract content was determined by the weight variation in the tea leaves (ISO 9768:1994). The contents of eight catechins (ISO 14502-1/2:2005) and caffeine (ISO 20727:1995) were determined by high-performance liquid chromatography (SIL-20AC; SHIMAZU Company, Kyoto, Japan). The eight catechins included epicatechin gallate (ECG), epigallocatechin gallate (EGCG), gallocatechin (GC), epigallocatechin (EGC), catechin (C), epicatechin (EC), gallocatechin gallate (GCG), and catechin gallate (CG). The total polyphenols (ISO 14502-1/2:2005) and free amino acids (GB8314-2013) were determined using UV spectroscopy (i3; Hanon company, Jinan, China).

### 4.3. Sensory Evaluation

All tea samples were examined and scored by seven tea tasting panelists from the Tea Research Institute, Zhejiang University, China. The sensory evaluation procedure was followed according to the National Standard of the People’s Republic of China (GB/T 23776-2018). The total score of sensory evaluation is 100, with 25% allocated for dry tea appearance, 10% for liquor color, 25% for aroma, 30% for taste, and 10% for infused leaves. The procedure was performed as follows: the appearance of dry tea samples was first evaluated and scored; second, 150 mL of boiled water was added to a cup with 3 g of dry tea sample. After covering for 4 min, the tea infusion was filtered, and the aroma, taste, and color of the tea infusion were examined, as was the appearance of wet leaves. Finally, the total score was calculated [29].

### 4.4. UV Spectrum Sampling

A microplate reader (Synergy H1, BioTek company, Winooski, VT, USA) was used for collecting UV absorption spectra of tea samples. The scanning range was 230–380 nm, and the scanning interval was 1 nm. The sample preparation was the same as in Section 4.3. Tea infusion was diluted 20 times for UV spectroscopy sampling. We added 500 μL tea liquid onto a quartz microplate to obtain the absorption intensity. The UV spectrum provided the absorption peak and the absorption intensity of the absorption spectroscopy.

### 4.5. Color Difference Sampling

The color difference of the tea infusion was determined on an UltraScan VIS spectrophotometer. (CIE; HunterLab Company, Reston, VA, USA). We set the mode type as TTRAN-total transmission. The ultraviolet filter was nominal. Chalkboard and whiteboard were used for color correction. The sample preparation was the same as described in Section 4.3. We added 30 mL tea infusion into cuvette to obtain parameters of colors coordinate according to CIE L *, a *, b *, where L represents the brightness of the samples, where 0–100 represents the color from black to white; a is the red-green color, where a positive value indicates red and a negative value represents green; and b is the yellow-blue color, where a positive value indicates yellow and a negative value represents blue. ΔE and ΔE cmc represent the total color difference and ellipse ΔE formula, respectively. When ΔE is between 0 and 1, the color difference is indistinguishable to the naked eye. If ΔE is between 1 and 2, the human eye will slightly detect the difference. If ΔE is between 2 and 3, the color difference can be distinguished. Once ΔE reaches 3.5 to 5, the color difference is evident. When ΔE is above 5, the colors appear as two different colors. Distilled water was used as the blank sample. Thus, five features of color differences were obtained: ΔL, Δa, Δb, ΔE, and ΔE cmc. Measurements were recorded at a room temperature of 25 ± 1 °C [17].

### 4.6. Data Analysis

PCA is a commonly used dimension reduction method. It exports a few principal components from the original variables while retaining as much information as possible about the original variable. It can effectively reduce the dimensions of the original variables. Graphs represent the relationship between variables, which improve the efficiency of analysis [16,17]. SVM is a supervised learning classifier used for binary classification of data that is able to learn in a high-dimensional feature space with fewer training data. SVM analysis has been widely used in tea classification by spectroscopy [30]. RF is a classifier that uses multiple trees to train and predict a sample. The RF classifier has performed well in the prediction of unknown tea samples [31]. In this study, we compared the classification and prediction effects of SVM and RF on tea samples of different grades using single and composite models. For each sample, we used triplicates for the detection of chemical component and 12 repetitions for digital evaluation. UV spectroscopy had 72 × 151 signals. The color difference had 72 × 5 signals, and the combined data had 72 × 4 signals. The python sklearn and pandas library (python 3.7) were used to perform data analysis, including PCA, SVM, and RF [17].

## 5. Conclusions

In this study, the UV spectrum and color difference were combined to classify the grade of Huangshan Maofeng tea. The combination of these two factors, instead of sensory evaluation, outputs digital results to ensure the objectivity of the tea grade identification. Compared to complex detection of chemical components, this method does not require pretreatment—the use of the tea infusion is sufficient—thereby considerably accelerating the efficiency with high accuracy rates. To classify Huangshan Maofeng tea, the combined signal was more accurate than the single-signal UV spectrum and the single-signal color difference. The combined signal’s PCA plot effectively separated samples according to grade, and showed a regular distribution of decreasing quality in the clockwise direction. In terms of discrimination of quality, the RF model was more suitable. The training accuracy was 1 for all signals and the testing accuracy of the combined signal was 1. With the advantages of fast detection and accurate classification, UV spectrum and color difference combined the with RF model can be used as an effective method for tea quality identification. In the future, it can be applied to other kinds of teas, such as black tea, dark tea, and so on.

## 6. Patents

An application for patent has been submitted to the National Intellectual Property Administration, PRC. The patent application number is 202010744113.8.

## Figures and Tables

**Figure 1 molecules-25-04665-f001:**
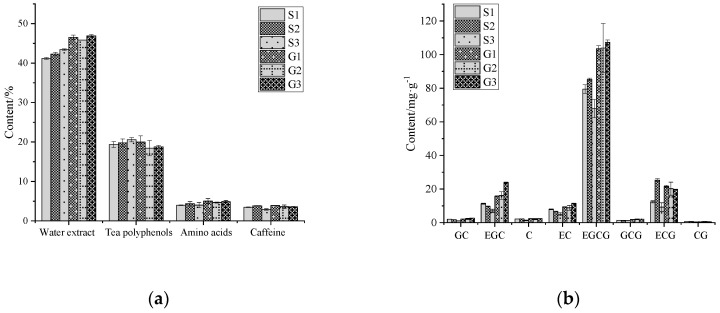
Histogram of mean chemical composition content in tea: (**a**) the average chemical content of each grade of tea and (**b**) the average catechin monomer content of each grade of tea. ECG: epicatechin gallate, EGCG: epigallocatechin gallate, GC: gallocatechin, EGC: epigallocatechin, C: catechin, EC: epicatechin, GCG: gallocatechin gallate, CG: catechin gallate.

**Figure 2 molecules-25-04665-f002:**
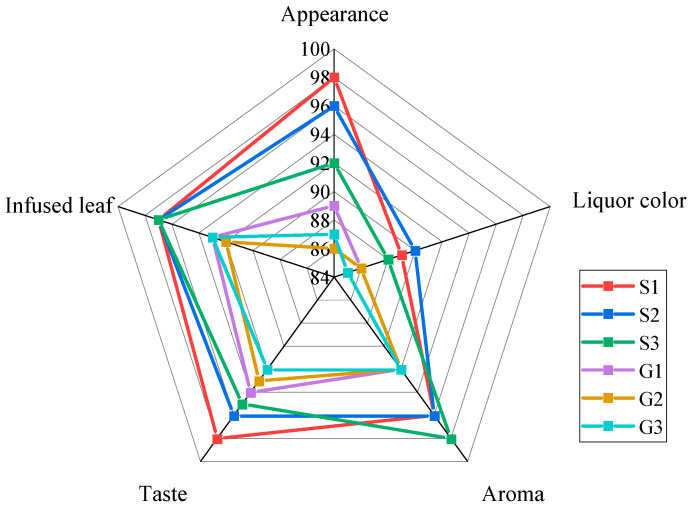
Sensory evaluation scores of each grade.

**Figure 3 molecules-25-04665-f003:**
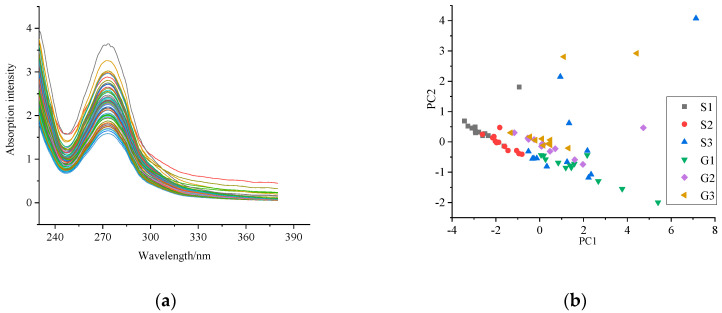
UV spectrum of samples: (**a**) the original UV spectra of samples and (**b**) the PCA plot of the UV spectrum of each grade.

**Figure 4 molecules-25-04665-f004:**
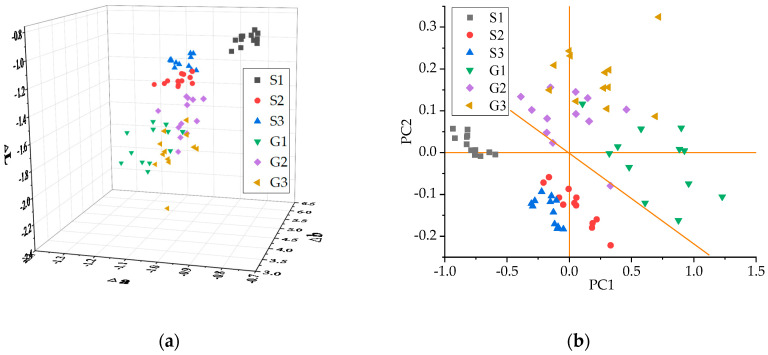
Color difference of samples: (**a**) the three-dimensional diagram of the tea color difference and (**b**) PCA plot of the color difference of each tea grade.

**Figure 5 molecules-25-04665-f005:**
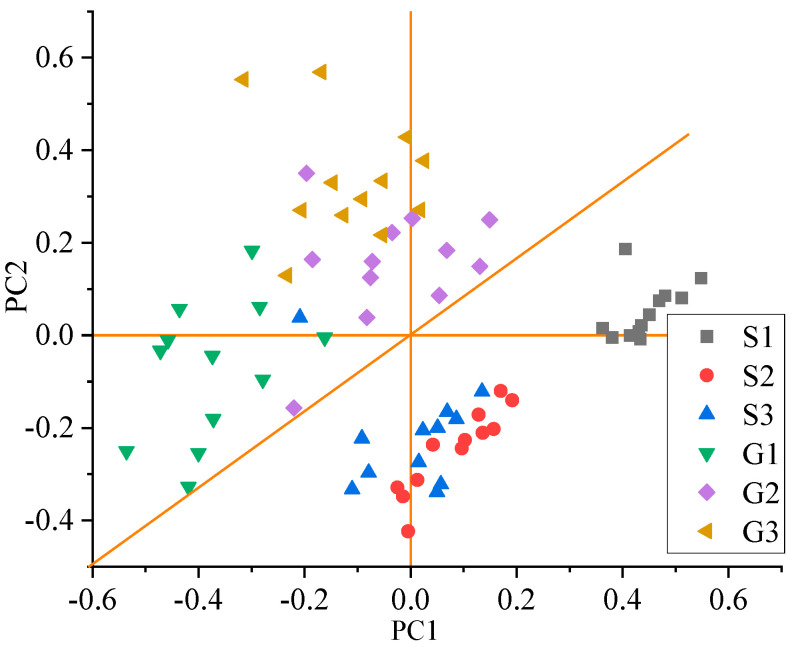
PCA plot of the combined signals of each tea grade.

**Table 1 molecules-25-04665-t001:** The classification results based on UV spectrum, color difference, and combined signal using the three models.

Model	Data	S1	S2	S3	G1	G2	G3	Training Accuracy	Testing Accuracy
SVM	UV spectrum	1	1	1	1	1	1	1	0.750
	Color difference	1	0.700	1	0.909	0.800	0.818	0.875	1
	Combined signal	1	1	1	1	0.700	0.818	0.922	0.875
RF	UV spectrum	1	1	1	1	1	1	1	0.500
	Color difference	1	1	1	1	1	1	1	0.875
	Combined signal	1	1	1	1	1	1	1	1
SVM + RF	UV spectrum	1	1	1	1	1	1	1	0.500
	Color difference	1	1	1	1	1	1	1	0.875
	Combined signal	1	1	1	1	1	1	1	1

**Table 2 molecules-25-04665-t002:** The optimum parameter based on UV spectrum, color difference, combined signal using the three models.

Model	SVM		RF	SVM + RF		
Parameter	C	Gamma	*n*	C	Gamma	*n*
UV spectrum	21.800	27.210	82	5.556	5.556	69
Color difference	27.483	27.210	35	22.222	11.111	12
Combined signal	33.033	5.442	18	38.889	44.444	99

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
