# Peer review of "Rapid Identification of Different Grades of Huangshan Maofeng Tea Using Ultraviolet Spectrum and Color Difference"

_molecules, 2020, doi:10.3390/molecules25204665_

Round 1
Reviewer 1 Report
In the submitted manuscript title "Rapid Identification of Different Grades of Huangshan Maofeng by Ultraviolet Spectrum and Chromatic Aberration" reflect the content and emphasize the paper's interest and significance. The suggestion to the authors is to add the word "tea" as well (Rapid Identification of Different Grades of Huangshan Maofeng tea by Ultraviolet Spectrum and Chromatic Aberration).
Abstract /Rewrite it: focus more on how your research has contributed to knowledge gaps; describe research limitations for future research and restate your major findings.
Introduction /The introduction provides a good, generalized background of the topic that quickly gives the reader an appreciation of the wide range of applications for this technology. The objective of this study is clearly defined.
R&D /The results presented in the manuscript are interesting.
Improve the quality of attached Figures (higher resolution). Below each figure, it is necessary to clarify the abbreviations (legend) that appear on them.
M&M /The methods used in this study are standard and appropriate.
4.5. Chromatic Aberration Sampling / Explain colour parameters in more detail (range of values, positive/negative values and their meaning). It is necessary to provide a reference for the specified method.
4.6. Data Analysis/ Explain in more detail the PCA method used (indicate both the software and the number of repetitions of analyses); provide a reference for the method.
Conclusions /Given the scope of the results presented, it is necessary to improve this section. Please rewrite it clearly stating the facts; focus more on how your research has contributed to knowledge gaps; describe research limitations for future research and restate your major findings. Please add scientific and practical significance of the selected method.
References /The literature cited is relevant to the study.
Overall, English language is acceptable.
It was a pleasure to read this manuscript. I wish the author of the best.
Reviewer 2 Report
Type of manuscript: Article
Title: Rapid Identification of Different Grades of Huangshan Maofeng bybUltraviolet Spectrum and Chromatic Aberration
Journal: Molecules
Ref: molecules-934961
This paper describes new analytical methods for the Rapid Identification of Different Grades of Huangshan Maofeng by Ultraviolet Spectrum and Chromatic Aberration. The paper is original and it can be consider as relevant in analytical chemistry methods.
Interesting analytical methodology for the quality control of teas in less complex laboratorios
However I have some comments about this paper
The manuscript is well presented, with methodological rigor, the multivariate analysis could be improved in terms of the presentation of figures and tables so that they are better understood by the future reader.
Explain in more detail the chromatic aberration methodology.
Correct minor details of the English language.
As conclusion, I consider that this paper can be published in MOLECULAS after with minor corrections.
Reviewer 3 Report
To the Authors,
in manuscript (molecules-934961-peer-review-v1), titled "Rapid Identification of Different Grades of Huangshan Maofeng by Ultraviolet Spectrum and Chromatic Aberration", authors Huang et al. "proposed a quick method that combined ultraviolet (UV) spectrum and chromatic aberration to classify [Huangshan Maofeng] tea grade". They concluded that the "method brings great convenience to grade identification of tea."
Authors argue that Huangshan Maofeng tea's classification based on sensory assessment "needs professional knowledge, [and is] confusing and troublesome for consumers; it is urgent to construct a simple, rapid and reliable method to judge the quality of tea." Their proposed method could contribute to this goal. The paper reports a relatively straightforward experiment that is reproducible elsewhere. It is an interesting study. Seemingly, it as real-life application. Manuscript's English requires revision (to convey context, hypothesis and arguments). A number of suggestions are made to the text. Aren't there any disadvantages in the method?
Specific suggestions are made directly in the PDF using Adobe Reader DC tools. Some (more general issues) are highlighted next:
-add "tea" in the title as "Huangshan Maofeng tea" (eventually do the same in the text); revise the title.
-substitute the expression "chromatic aberration" (it is cumbersome and, to my knowledge, unused in the field of food science and technology), by " colorimetry" (since authors are using Lab color system). Also, clarify which color system CIE or Hunter (L.266) because they're diff.
-no need to refer to "human sensory evaluation" (and the like), "sensory evaluation" is enough and more common. Moreover, As in the case of other regulated international food products and beverages, like wine, coffee or cheese, supposedly this involves trained panelists that compose official panels and work at certification bodies? It complements physicochemical parameters of quality. However, the proposed method requires a "microplate reader" and "UV/Vis spectrophotometer".
-consider using "combined" instead of "fused"
-not clear if the "traditional", practiced grading of tea is made using fresh or processed leafs or their infusion? if based on the assessment of leafs how does this study compare since it is based on the infusion? -the stats analysis, namely the PCA "part" needs completion of description the "analytical" procedures (in the Material and Methods) as well as of the results (e.g. add biplots and eigenvectors/eigenvallues/etc eventually as suppl material). Likewise, SVM, RF and SVM-RF requires much more info.
-more info about the sensory evaluation needed: Number of panelists? what about the scales used?
-doesn't "all samples belonged to lower phenolic" (L. 81-82) bias the study?
-check in-text citation format (e.g. L. 194)

Author Response
Please see the attachment.

This manuscript is a resubmission of an earlier submission. The following is a list of the peer review reports and author responses from that submission.